# Comparative Genomic Analysis of a Thermophilic Protease-Producing Strain *Geobacillus stearothermophilus* H6

**DOI:** 10.3390/genes14020466

**Published:** 2023-02-11

**Authors:** Ruilin Lai, Min Lin, Yongliang Yan, Shijie Jiang, Zhengfu Zhou, Jin Wang

**Affiliations:** 1College of Life Science and Engineering, Southwest University of Science and Technology, Mianyang 621000, China; 2Key Laboratory of Agricultural Microbiome (MARA), Biotechnology Research Institute, Chinese Academy of Agricultural Sciences, Beijing 100081, China

**Keywords:** comparative genomics, genome sequencing, *G. stearothermophilus* H6, thermostability, protease

## Abstract

The genus *Geobacillus* comprises thermophilic gram-positive bacteria which are widely distributed, and their ability to withstand high temperatures makes them suitable for various applications in biotechnology and industrial production. *Geobacillus stearothermophilus* H6 is an extremely thermophilic *Geobacillus* strain isolated from hyperthermophilic compost at 80 °C. Through whole-genome sequencing and genome annotation analysis of the strain, the gene functions of *G. stearothermophilus* H6 were predicted and the thermophilic enzyme in the strain was mined. The *G. stearothermophilus* H6 draft genome consisted of 3,054,993 bp, with a genome GC content of 51.66%, and it was predicted to contain 3750 coding genes. The analysis showed that strain H6 contained a variety of enzyme-coding genes, including protease, glycoside hydrolase, xylanase, amylase and lipase genes. A skimmed milk plate experiment showed that *G. stearothermophilus* H6 could produce extracellular protease that functioned at 60 °C, and the genome predictions included 18 secreted proteases with signal peptides. By analyzing the sequence of the strain genome, a protease gene *gs-sp1* was successfully screened. The gene sequence was analyzed and heterologously expressed, and the protease was successfully expressed in *Escherichia coli*. These results could provide a theoretical basis for the development and application of industrial strains.

## 1. Introduction

Thermophilic microorganisms are microbes that can grow at 41~122 °C, and their optimal growth temperature is 45~80 °C. Thermophilic ecological environments are distributed at sites including volcanoes, geothermal areas (terrestrial, underground and marine hot springs), compost, oil reservoirs and other extreme high-temperature areas on Earth [1]. Many thermophilic microorganisms play an important role in biotechnology and have major commercial applications in industrial production [2]. For example, they can produce a variety of thermostable enzymes [3], generate biofuels by degrading agricultural wastes [4], and they show a special leaching capacity for certain minerals [5] and a bioremediation capacity [6].

*Geobacillus* was separated from *Bacillus* in 2001 by T.N. Nazina and others as a new genus of bacteria [7]. It is mainly composed of aerobic or facultative anaerobic bacteria with the ability to form endophytic spores and is a typical thermophilic microbial group [8]. The genus is widely distributed and is found in natural environments such as oil fields, hot springs, volcanic vents, dairy plants, food processing, compost and other high-temperature environments [9]. The ability of *Geobacillus* to grow at high temperatures makes it suitable for various applications in biotechnology and industrial production. R.E. Cripps and others used metabolic engineering to transform two *Geobacillus* thermophilic bacteria to obtain a strain that can efficiently produce ethanol [10]. *Geobacillus* species can secrete extracellular polysaccharides and bacteriocins and show bioremediation properties [11]. They can be a source of many thermostable enzymes, such as xylanase, lipase, protease and amylase [12,13,14,15,16]. These thermostable enzymes play an important role in the commercial production of detergent, the brewing industry and the food industry [17].

Proteases are enzymes that catalyze the cleavage of protein peptide bonds and are widely found in animals, plants and microorganisms. They present important industrial applications and are widely used in the detergent, leather, medicine and other industries. Their output accounts for more than 65% of the enzyme preparation market [18]. Microbial proteases are the most important source of commercial proteases. *Bacillus* species can secrete a variety of proteases; this genus is the most important source of commercial production and is of great significance in commercial protease production [19]. Some *Geobacillus* strains have been proven to be capable of producing proteases [20]. Proteases isolated from *Geobacillus* can also adapt to high-temperature environments and can be used in a variety of industrial production environments [21].

*Geobacillus* is an important source of thermostable enzymes and thermophilic proteases. This study analyzed a thermophilic bacterial strain, *G. stearothermophilus* H6, isolated from hyperthermophilic compost in Beijing. After sequencing and analyzing the whole genome of *G. stearothermophilus* H6, we further analyzed its gene functions using bioinformatics tools.

## 2. Materials and Methods

### 2.1. Sample Collection, Strain Isolation and Culture

The samples used in this study were hyperthermophilic composting soil samples from Beijing. To isolate thermophilic bacteria, LB and R2A liquid culture media were used, and 5 g soil samples were added to 50 mL liquid culture media and were then incubated in a water bath at 80 °C for 48 h. Then, 500 μL of enrichment solution was added to 50 mL of new liquid medium for further culture in an 80 °C water bath, and the enrichment solution was collected three times. The enrichment solution was diluted to 10^−1^, 10^−2^, 10^−3^ and 10^−4^ with ddH_2_O, and the dilutions were spread on corresponding agar plates and cultured in an 80 °C incubator. A colony was selected for 16S rRNA sequencing, and the bacteria were preserved. The strain was isolated from an R2A agar plate.

### 2.2. Genomic DNA Extraction and Sequencing of G. stearothermophilus H6

*G. stearothermophilus* H6 was cultured in LB medium at 60 °C overnight. The bacterial solution was centrifuged at 4000 r/min at 4 °C for 10 min, the supernatant was discarded, and the bacterial cells were collected. Total bacterial DNA was extracted according to the operating instructions of a bacterial genomic DNA isolation kit (Mei5 Biotechnology Co. Ltd., Beijing, China), and the concentration and quality of the DNA were assessed with a NanoDrop 2500 system (OD_260_/OD_280_ = 1.8–2.0, ≥10 µg). The total DNA of the extracted samples was stored on dry ice and sent to Biomarker Technologies to complete the sequencing analysis.

### 2.3. Phylogenetic Tree and Comparative Genomic Analysis of 16S rRNA of G. stearothermophilus H6

Genomic DNA was extracted and purified with a commercial bacterial genomic DNA isolation kit. The 16S rRNA gene was amplified with the universal bacterial primers 27F and 1492R. Preliminary sequence analysis of the 16S rRNA gene was conducted using the NCBI database, and strains with high homology in the NCBI database were selected for phylogenetic tree analysis. The corresponding phylogenetic tree was constructed by using MEGA 6.0 [22] software and the maximum likelihood method (ML). The evolutionary tree was constructed based on the bootstrap values of 1000 repeats.

Five homologous strains were selected, their basic information was compared with that of *G. stearothermophilus* H6, and the average nucleotide identity (ANI) was calculated (www.ezbiocloud.net/tools/ani (accessed on 15 October 2022)). Using the Mauvealigner algorithm of Mauve 2.4.0 [23], the whole-genome sequence of *G. stearothermophilus* H6 and the whole-genome sequence of the reference located close to the source strain were analyzed for collinearity.

### 2.4. Detection of Protease Activity of G. stearothermophilus H6

*G. stearothermophilus* H6 bacterial solution (2.5 μL) was cultured to the middle logarithmic phase spotted onto a skim milk plate, and the results were compared with *Bacillus velezensis*, *Bacillus subtilis* and *E. coli* BL21 (DE3). The four strains were cultured at both 37 °C and 60 °C. After different culture times, the transparent circles that appeared were observed to preliminarily judge the protease production ability of the strains.

### 2.5. Whole-Genome Sequencing and Analysis

The original genome data were filtered and more than 2 kb of reads were retained. Hifiasm v0.16.0 2 [24] software was used to assemble the filtered reads. Circulator v1.5.5 software was used to cyclize and adjust starting sites. Pilon v1.22 software was used to further correct errors using second-generation data, and a genome with higher accuracy was obtained for subsequent analysis. Prodigal v2.6.3 [25] was used to predict coding sequences (CDSs) in the genome of the strain, and genome information obtained by assembly and prediction, such as information on tRNAs, rRNAs, repeat sequences, GC contents and gene functions, was used to draw a circular genome map with the software Circos v0.66 [26].

We used software to predict repeat sequences, rRNAs, tRNAs, CRISPRs, and gene islands in the genome. Gene function annotation was mainly based on protein sequence comparison, performed by comparing the gene sequences in each database. The predicted gene sequences were compared with eggNOG, KEGG, Swiss-Prot, TrEMBL, Nr, GO, Pfam and other databases to obtain gene function annotation results.

### 2.6. Analysis of G. stearothermophilus H6 Protease

General databases such as eggNOG, KEGG, Swiss-Prot, TrEMBL, Nr, GO, and Pfam were used to predict the distribution of proteases in strain *G. stearothermophilus* H6, and the software SignalP v4.0 [27] was used to predict protein signal peptides and specific signal peptide excision sites for further analysis. At the same time, TMHMM v2.0 [28] was used to predict the transmembrane domains of the protease. The similarity of the protease gene between *G. stearothermophilus* H6 and the other two homologous strains was compared with gene sequences in the NCBI database.

### 2.7. Heterologous Expression of the G. stearothermophilus H6 Protease Gene in E. coli

After screening the protease genes of *G. stearothermophilus* H6, the *GE001730* gene was selected and named *gs-sp1*. The thermophilic protease gene fragment was then inserted into the pET22b vector digested by *Hin*d III and *Xho* I using seamless cloning and recombination technology.

PCR technology was used to amplify the gene (95 °C for 5 min, 1 cycle; 95 °C for 30 s, 58 °C for 30 s and 72 °C for 30 s, 33 cycles; 72 °C for 10 min, 1 cycle) using the following primers: F (5′-tcgagctccgtcgacaagcttATCTTCCCTCATATGAGTATAGGA-3′) and R (5′- gtggtggtggtggtgctcgagGCGCTGCAGCAGTTGCTC-3′). The PCR products were purified with a gel recovery kit (Mei5 Biotechnology Co. Ltd., Beijing, China). The PCR products were cloned into the expression plasmid pET-22b using the ClonExpress Ultra One Step Cloning Kit (Vazyme, Ninjing, China) and then transformed into *E. coli* BL21 (DE3) cells for further screening and verification.

The constructed *E. coli* BL21/pET22b/gs-sp1 strain was selected on Amp-resistant plates and cultured overnight at 37 °C. A single colony was picked and cultured overnight at 220 rpm in 20 mL liquid medium at 37 °C. Then, 2% of the volume was transferred to 20 mL LB medium, and culture was continued until reaching OD_600_ = 0.5~0.7. Thereafter, 0.1 mM IPTG was used to induce protein expression at 37 °C for 4 hours, and a 5 μL aliquot was spotted on a skim milk plate. The wild-type *E. coli* BL21 and *E. coli* BL21/pET22b strains were used as controls, and the transparent circles produced by the bacteria were observed.

## 3. Results

### 3.1. Phylogenetic Analysis of G. stearothermophilus H6 16S rRNA

To clarify the taxonomic position of *G. stearothermophilus* H6, 16S rRNA analysis was used. *G. stearothermophilus* H6 is a thermophilic bacterium obtained from hyperthermophilic composting soil by culture in R2A medium and screening based on 80 °C culture. According to 16S rRNA gene sequence analysis, the strain is a bacterium of the genus *Geobacillus* showing the highest homology with *G. stearothermophilus* B5; therefore, the strain was named *G. stearothermophilus* H6. The 16S rRNA gene of *G. stearothermophilus* H6 presented the highest similarity with *G. stearothermophilus* B5 (99.97%), *G. stearothermophilus* D1 (99.97%), *G. stearothermophilus* DSM 458 (99.93%), *G. stearothermophilus* DG-1 (99.93%), *G. stearothermophilus* IFO 12550 (99.79%) and *G. stearothermophilus* 10 (99.78%). A phylogenetic analysis of the 16S rRNA gene with a tree constructed based on the maximum likelihood (ML) method showed that *G. stearothermophilus* H6 was a member of the genus *Geobacillus* (Figure 1).

### 3.2. Comparative Genome Analysis of G. stearothermophilus H6

To compare the differences between *G. stearothermophilus* H6 and the five most closely related strains (*G. stearothermophilus* DSM 458, *G. stearothermophilus* 10, *G. stearothermophilus* DG-1, *G. stearothermophilus* D1 and *G. stearothermophilus* B5), the genomic characteristics of the six strains were statistically analyzed, and the results are shown in Table 1. The average nucleotide homology (ANI) value indicates the similarity between the sequences of the conserved regions of two genomes and allows the genetic relationships between them to be analyzed. According to the whole-genome information of these six strains, the ANI of the genome of *G. stearothermophilus* D1 was highest (97.65%), showing good homology (Table 1). The genome sizes and GC contents of the six strains were similar, with genome sizes ranging from 2.97–3.65 Mb and GC contents from 51.66–52.6%.

Based on 16S rRNA phylogenetic tree analysis, the genomes of *G. stearothermophilus* B5 and *G. stearothermophilus* 10, which show close homologous relationships with *G. stearothermophilus* H6, were selected. The software Mauve 2.4.0 was used to perform genome synteny analysis and quickly analyze whether large-segment sequence rearrangements existed between genomes. The squares with similar colors represent highly homologous assembly regions of the two genomes. Figure 2 shows that *G. stearothermophilus* H6 and *G. stearothermophilus* 10 had poor synteny, with many gene rearrangements, such as insertions, deletions, inversions and translocations, between them. For example, compared with *G. stearothermophilus* 10, there was a gene deletion at 1,177,537–1,516,081 bp in *G. stearothermophilus* H6, and an inversion occurred at 1,604,138–1,649,959 bp in *G. stearothermophilus* H6. *G. stearothermophilus* H6 presented good synteny with *G. stearothermophilus* B5, but there were also some gene rearrangements between them, such as deletions and inversions. For example, compared with *G. stearothermophilus* B5, a gene inversion occurred at 1,786,619–1,868,252 bp in *G. stearothermophilus* H6, and a deletion occurred at 2,464,988–2,718,550 bp in *G. stearothermophilus* H6 (Figure 2).

### 3.3. Detection of Protease Activity in G. stearothermophilus H6

The protease hydrolysis activity of *G. stearothermophilus* H6 was observed by the skimmed milk plate method and compared with that of *B. velezensis*, *B. subtilis* and *E. coli* BL21(DE3). The four strains were cultured at both 37 °C and 60 °C, and the results showed that *G. stearothermophilus* H6 could produce transparent circles that became larger with increasing culture time (Figure 3). When cultured at 37 °C, *B. velezensis*, *B. subtilis* and *G. stearothermophilus* H6 produced transparent circles, with *B. velezensis* producing the strongest degradation results. When cultured at 60 °C, *G. stearothermophilus* H6 produced the largest transparent circle, and the other three strains did not produce a transparent circle. Thus, *G. stearothermophilus* H6 produces extracellular proteases that function under high temperature and show a good effect.

### 3.4. Overview of the Genome Assembly and Whole Genome of G. stearothermophilus H6

Based on the specificity of the high-temperature tolerance of *G. stearothermophilus* H6, the whole genome of the strain was sequenced to further explore the specific coding genes associated with its high-temperature tolerance. Gene prediction was carried out with Prodigal v2.6.3 software, and a genome completion map was obtained through assembly and construction. The size of the genome sequence of *G. stearothermophilus* H6 was 3,054,993 bp, and the average GC content was 51.66%. It was predicted that there were 3750 coding genes with an average length of 814 bp in the genome. The results of noncoding RNA prediction showed that the genome contained 30 rRNAs and 91 tRNAs and had 17 CRISPR regions and 24 gene islands (Table 2).

Based on the genome information obtained by assembly and prediction, such as information on tRNAs, rRNAs, repeat sequences, GC contents and gene functions, Circos v0.66 software was used to obtain the circular genome map (Figure 4).

The amino acid sequence of *G. stearothermophilus* H6 was compared with the Nr database, and the corresponding species information was obtained from the annotation database. Through BLAST searches comparing the protein sequences of genes with the Nr database, the most similar sequences in the Nr database could be found. The corresponding annotation information of the sequences was the annotation information of the corresponding gene in the genome sequence. A total of 3699 genes were annotated.

BLAST comparisons of the protein-encoding gene sequences of the whole genome were performed against the eggNOG database, and a database of the results was generated. The database was frequently used to classify and annotate genes of newly sequenced genomes. The annotation information and classification information in the genome corresponded to the gene sequences of the sequenced genome. A total of 3186 genes were annotated in the database.

The amino acid sequences of *G. stearothermophilus* H6 were subjected to BLAST searches in the KEGG database to assemble databases of the biological pathways related to diseases, drugs and chemical substances in the genome. The strain has 1798 genes in the KEGG database.

The prediction results were annotated in the GO database. The number of genes dominated by GO functional classifications mainly included the highest-level functional nodes: cellular component, molecular function and biological process. A total of 2686 genes were predicted in the database (Figure 5).

### 3.5. Analysis of G. stearothermophilus H6 Protease

The predicted gene sequences were compared with eggNOG, GO, KEGG, Nr, Pfam, Swiss-Prot, TrEMBL and other general databases to obtain gene functional annotation results. Approximately 141 proteases were predicted, which accounted for approximately 3% of the total encoded proteins. The predicted proteases mainly consisted of serine proteases and metalloproteinases and a few cysteine proteases, aspartic proteases and threonine proteases, accounting for 19%, 27%, 1%, 2% and 1% of the predicted proteases, respectively. Other proteases could not be characterized (Table 3).

Secretory proteins are proteins secreted from the cells of living microorganisms. Through the prediction and analysis of signal peptides and secretory proteins in the genome, approximately 161 proteins with signal peptides were identified, which could form secretory proteins. The predicted proteases included 18 secretory proteins with signal peptides, and the serine and metalloproteinases included 5 secretory proteins with signal peptides.

The 18 secreted proteases were analyzed and compared with the proteases of the *G. stearothermophilus* B5 and *G. stearothermophilus* 10 genomes. Among them, GE000377 was not predicted to show homologous proteins in *G. stearothermophilus* B5 but showed higher homology with the proteins of *G. stearothermophilus* 10 (99.24%); GE003445 was not predicted to show homologous proteins in *G. stearothermophilus* 10 but presented homologous proteins with *G. stearothermophilus* B5 (76.00%). The homologous proteins encoded by the GE002130, GE003446, GE003450 and GE003532 genes in the two homologous strains exhibited low similarity. These proteins may be encoded by genes unique to *G. stearothermophilus* H6. Among the 18 proteases with signal peptides, most of the signal peptides were removed by the type I signal peptidase SP (Sec/SPI), and only the signal peptides of the GE000438 and GE002405 genes were removed by the type II signal peptidase LIPO (Sec/SPII) (Table 4).

The prediction and analysis of transmembrane helix structures in the genome indicated that approximately 841 proteins had transmembrane helix structures. Among these proteins, 46 proteases had transmembrane helix structures, while serine proteases, metalloproteinases and aspartic proteases had 12, 16 and 3 transmembrane helix structures, respectively.

### 3.6. Construction of GS-SP1 Protein Expression Vector and Verification of Secreted Proteases

The GS-SP1 protease gene was cloned into the pET22b expression vector, which contains the signal peptide *pelB* upstream of multiple cloning sites. Then, the constructed pET22b/gs-sp1 plasmid was transformed into *E. coli* BL21 (DE3) (Figure 6a), and wild-type *E. coli* BL21 and *E. coli* BL21/pET22b were used as controls. Five microliters of bacterial liquid culture was spotted onto a skimmed milk plate, and culture was performed at 37 °C for 24 h to observe transparent circle development. The results showed that *E. coli* BL21/pET22b/gs-sp1 could produce a transparent circle when induced by 0.1 mM IPTG, while the other two strains could not (Figure 6b). These results indicated that the GS-SP1 protein showed protease activity, and further investigation of this protein will be important for the exploitation of thermophilic proteases.

## 4. Discussion

In this study, we isolated a strain of *Geobacillus* from hyperthermophilic compost and named it *G. stearothermophilus* H6. Thereafter, 16S rRNA sequence analysis and comparisons showed that the strain presented the highest consistency with *G. stearothermophilus* B5 (99.97%). The skimmed milk plate experiment showed that *G. stearothermophilus* H6 could produce extracellular proteases with enzyme activity at high temperature. Whole-genome sequencing and genome annotation analysis revealed that *G. stearothermophilus* H6 produces a variety of enzymes with biotechnological significance, such as proteases, amylases and lipases. Thus, it may be an important source of thermophilic enzymes and has important research value.

*Geobacillus* is a genus of thermophilic Gram-positive bacteria belonging to Bacillaceae, including denitrifying bacteria, facultative anaerobes and obligate aerobic bacteria, which can grow at 45–80 °C [29]. The members of the genus can form endophytic spores, which can diffuse through global atmospheric circulation [30] and are widely distributed in environments such as in soil, hot springs, dairy plants or other food processing plants [8]. The chromosomes and plasmids of *Geobacillus* species exhibit significant genetic diversity. Bezuidt et al. [31] analyzed the pangenome of 29 genome sequences of *Geobacillus* sp. and found that the core genome was relatively small, mainly consisting of *Bacillus*-related genes, indicating that these bacteria originated from an ancestor of *Bacillus*; it contained a large number of dispensable genomes, which showed that *Geobacillus* spp. can achieve extensive genomic diversity through horizontal gene transfer, which is the key mechanism whereby *Geobacillus* spp. adapt to different environmental niches. For example, *G. stearothermophilus* obtained the *lac* operon through horizontal gene transfer, enabling it to survive in dairy products [32]. This feature provides a new way to produce thermostable enzymes for industrial use through the evolution of thermoadaptive directed enzymes, thus expanding the biotechnological application of *Geobacillus* spp. For example, *G. kaustophilus* HTA42 producing thermostable variants of rRNA methyltransferase was generated through thermal-adaptation-directed evolution [33]. *G. stearothermophilus* H6 shows potential as a host for whole-cell applications and a biological tool in evolutionary engineering.

The characteristics and distribution of proteases in the *G. stearothermophilus* H6 genome indicate that its proteases consist of serine proteases, metalloproteinases, cysteine proteases and aspartic proteases, and the proportion of proteases from PDB entries distributed in all *Bacillus* genomes is similar [34,35]. The exploration of the proteases of this strain may provide knowledge for the discovery of new potential proteases with various potential industrial applications. By analyzing *G. stearothermophilus* H6 genome proteases, we screened 18 proteases with signal peptides, selected the *gs-sp1* gene for heterologous expression, and successfully expressed the protease in *E. coli* BL21. Compared with the homologous strains *G. stearothermophilus* B5 and *G. stearothermophilus* 10, the gs-sp1 protease had higher homology. In addition, *G. stearothermophilus* H6 had unique protease genes, like the *GE003532* gene, which had low similarity among homologous strains. The *G. stearothermophilus* H6 genome also contains a variety of other enzymes which may have high thermal stability and broad application prospects in biotechnology applications. The ability of the thermophilic *Geobacillus* microorganisms to grow under high temperatures makes them a valuable resource for the development of new biotechnological applications [36]. They can be a source of many thermophilic enzymes, such as proteases, xylanases, amylases and lipases, and can be used for the synthesis of biofuels, such as bioethanol, isobutanol, biogas and biodiesel [37,38,39].

Currently, many species of *Geobacillus* are used to produce thermophilic enzymes either naturally or through the introduction of genetic engineering. Thermophilic enzymes are mainly used in biotechnology [40], including the food industry, detergent industry, leather industry, and medical industry [41]. The proteases isolated from *Geobacillus* sp. are extremely heat-resistant and can be used to improve the biodegradation of sewage sludge [42]. The optimum conditions for *Geobacillus* sp. YMTC 1049 to produce serine protease are 85 °C and pH 7.5 [43]. Due to the decreasing reserves of natural fossil fuels, the world needs to produce biofuels to develop alternative energy sources or fuels [8]. *Geobacillus* is used to biodegrade agricultural and industrial residues such as beet, soybean, barley, sugarcane, corn, sorghum and other biomass and produce biofuels through modern processes [44]. *G. stearothermophilus* has been employed to produce bioethanol using sucrose as a carbon source at approximately 70 °C, and the product yield is the same as that of yeast [45]. When *Geobacillus* strain AT1 is added to methanogenic sludge, it could effectively improve biogas production due to protease activity [46].

*G. stearothermophilus* is an important species of *Geobacillus* that can be employed as a source of various thermophilic enzymes and is widely used in a variety of biotechnology industries. Thermophilic enzymes produced by *G. stearothermophilus* SR74 α-amylase can be used in the papermaking, food and other industries [15]. *G. stearothermophilus* strain RM is used for the mass production of α-glucosidase at high temperature [47]. *G. stearothermophilus* PS11 can produce thermophilic and stable lipase under high temperature and alkali conditions, which is used for the production of biodiesel [17]. A protease cloned from *G. stearothermophilus* strain B-1172 has been used in the detergent and many other industries due to its catalytic domain and good activity [20]. *G. stearothermophilus* H6 isolated from hyperthermophilic compost can produce a protease with good activity at high temperature, which has broad application prospects in biotechnology applications.

## Figures and Tables

**Figure 1 genes-14-00466-f001:**
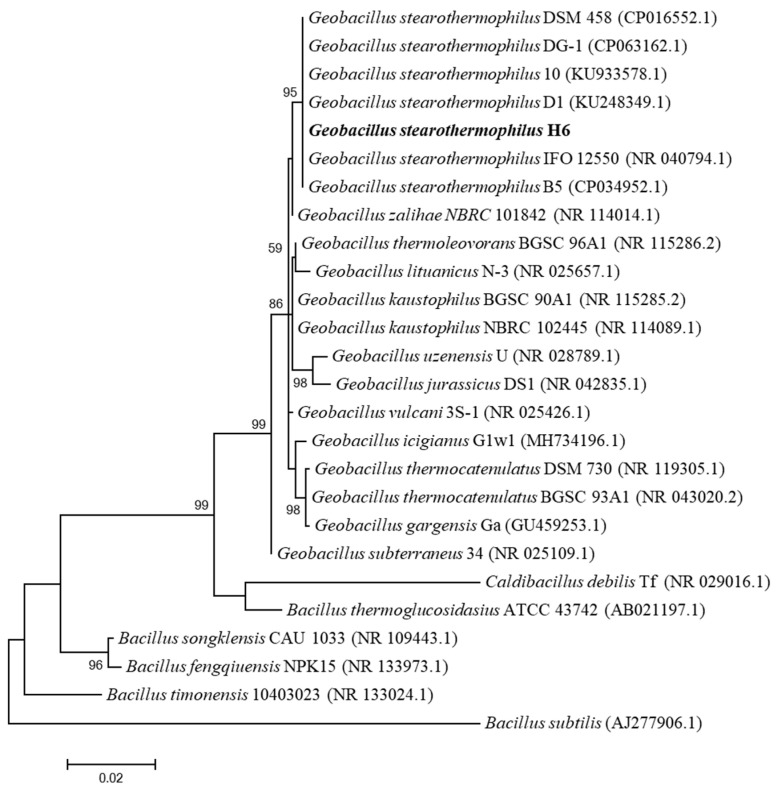
Phylogenetic tree of the 16S rRNA gene sequence based on maximum likelihood (ML). This tree shows the phylogenetic relationships between *G. stearothermophilus* H6 and closely related species. The GenBank registration number of the 16S rRNA gene sequence is shown in parentheses. Bootstrap test: the percentage is based on 1000 duplicates (bootstrap value > 70, indicating that the branch is reliable, and less than 50% will not be displayed).

**Figure 2 genes-14-00466-f002:**
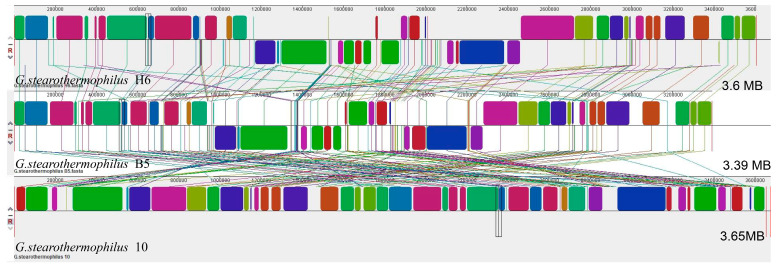
Genomic synteny analysis of *G. stearothermophilus* H6, *G. stearothermophilus* B5 and *G. stearothermophilus* 10. The same color block represents two genomes with high homology.

**Figure 3 genes-14-00466-f003:**
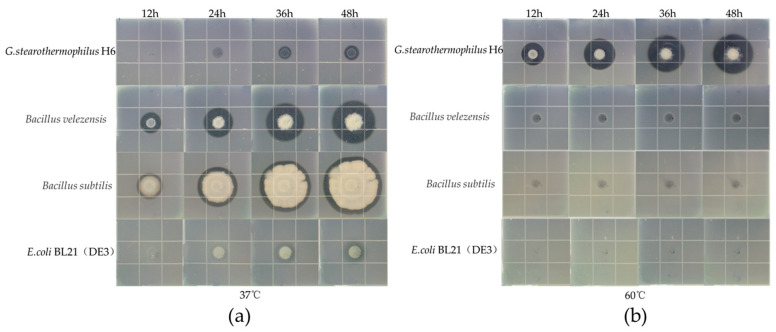
The experimental strains of bacteria were cultured on skimmed milk plates at 37 °C and 60 °C for 12 h, 24 h, 36 h and 48 h: (**a**) cultured at 37 °C; (**b**) cultured at 60 °C.

**Figure 4 genes-14-00466-f004:**
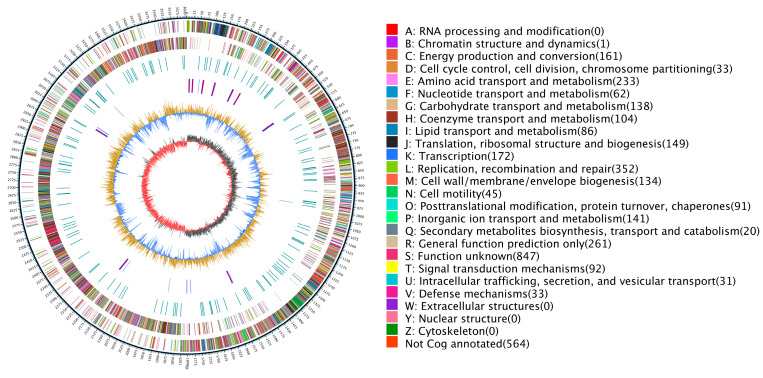
Circular genome map of *G. stearothermophilus* H6. Note: The outermost circle is a mark of genome size, each scale is 5 kb; the second and third circles are the gene on the positive and negative chains of the genome, respectively. Different colors represent different COG functional classifications; the fourth circle is the repeat sequence; the fifth circle is tRNA and rRNA, where blue is tRNA and purple is rRNA; the sixth circle is the GC content. The straw yellow part indicates that the GC content in this region is higher than the average GC content of the genome. The higher the peak value is, the greater the difference between the GC content and the average GC content is. The blue part indicates that the GC content in this region is lower than the average GC content of the genome; the innermost circle is GC skew. Dark grey represents the area where G content is greater than C, and red represents the area where C content is greater than G.

**Figure 5 genes-14-00466-f005:**
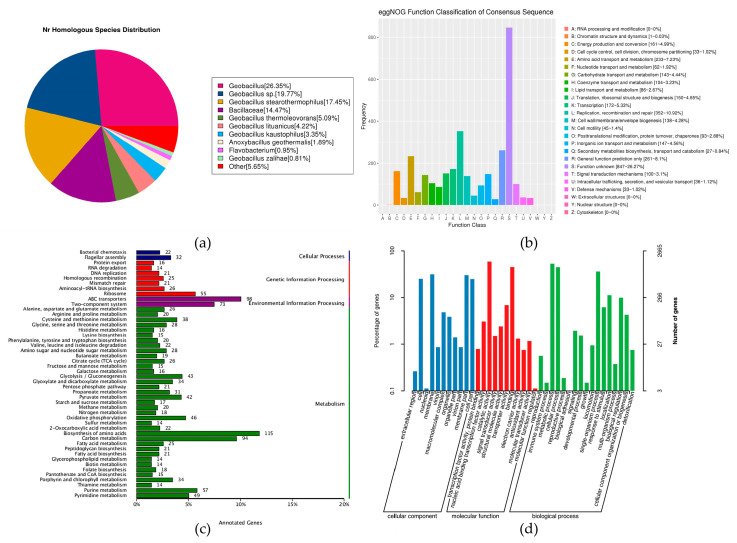
*G. stearothermophilus* H6 general protein prediction database: (**a**) Species distribution map of sequences compared to the Nr database: this map reflects the species distribution of the sequences compared to the Nr database. Different colors represent different species. (**b**) Statistical chart of the functional classification of eggNOG functional genes: the abscissa represents the classification content of eggNOG, and the ordinate represents the relative contents of the number of corresponding functional genes. (**c**) Statistical chart of KEGG annotation classifications: the ordinate represents the KEGG secondary classification, and the abscissa represents percentage. (**d**) Statistical chart of GO function annotation classifications: the abscissa represents the GO classification content, the left side of the ordinate represents the percentage of the number of genes, and the right side represents the number of genes. This figure shows the gene enrichment of each secondary GO function against the background of all genes, reflecting the status of each secondary function against this background.

**Figure 6 genes-14-00466-f006:**
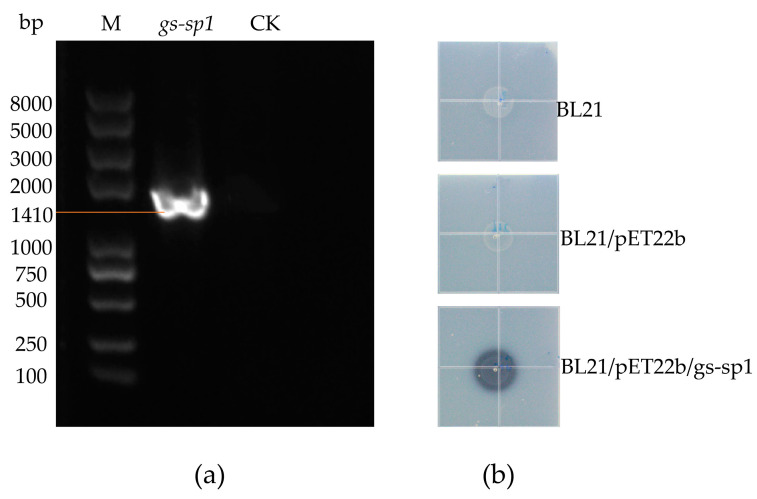
pET22b/gs-sp1 recombinant vector and protease activity observation: (**a**) PCR validation of the pET22b/gs-sp1 plasmid, M: Trans2K Plus DNA Marker; (**b**) after the strain was induced by 0.1 mM IPTG for 24 h, the transparent circle.

**Table 1 genes-14-00466-t001:** Comparison of basic characteristics of whole-genome sequences of strain H6 and other strains of *G. stearothermophilus*.

Type	Size (Mb)	GC%	Protein	Gene	Average Nucleotide Identity (ANI)	Isolation
*G. stearothermophilus* H6	3.6	51.66	3706	3750	-	hyperthermophilic composting
*G. stearothermophilus* B5 (CP034952.1)	3.39	52.5	3114	3446	97.35%	rice stack
*G. stearothermophilus* DSM 458 (CP016552.1)	3.47	52.1	3232	3614	96.35%	sugar beet juice from extraction installations
*G. stearothermophilus* DG-1 (CP063162.1)	3.51	52.5	3322	3627	97.07%	oilfield
*G. stearothermophilus* D1 (NZ_LDNU01000016.1)	2.97	52.2	2954	3482	97.65%	milk powder manufacturing plant
*G. stearothermophilus* 10 (CP008934.1)	3.65	52.6	3288	3473	89.56%	hot spring

**Table 2 genes-14-00466-t002:** General genomic characteristics of *G. stearothermophilus* H6.

Attribute	Value
Size (bp)	3,606,258
GC content (%)	51.66
Total genes	3750
RNA genes	121
rRNAs	30
tRNAs	91
Total repetitive sequence length (bp)	5600
CRISPR number	17
Number of genomic islands	24
Number of signal peptides	161
Transmembrane protein	841

**Table 3 genes-14-00466-t003:** Basic characteristics of proteases in the genome of *G. stearothermophilus* H6.

Types of Proteases	Number of Protease Sequences	Number of Signal Peptides	Number of Transmembrane Helixes
Serine protease	27	5	12
Metallopeptidase	38	5	16
Cysteine protease	1	0	0
Aspartic peptidase	3	0	3
Threonine peptidase	1	0	0
Other	71	8	15
Total	141	18	46

**Table 4 genes-14-00466-t004:** Basic characteristics and homology analysis of proteases secreted by *G. stearothermophilus* H6.

ID	Amino Acid Length	Average Molecular Weight (kDa)	Academic pl (pH)	Type of Signal Peptide	Homologous Protein and Identity	
					*G. stearothermophilus* B5	*G. stearothermophilus* 10
GE000011	451	51.00	9.64	SP(Sec/SPI)	WP_160269798.1 (36.20%)	ALA70391.1 (92.22%)
GE000377	263	30.03	5.84	SP(Sec/SPI)	-	ALA70722.1 (99.24%)
GE000438	150	16.72	10.42	LIPO(Sec/SPII)	WP_160268695.1 (97.33%)	ALA70781.1 (86.67%)
GE001521	341	36.35	9.47	SP(Sec/SPI)	WP_160269176.1 (90.62%)	ALA69075.1 (96.19%)
GE001730	453	49.85	6.01	SP(Sec/SPI)	WP_160269346.1 (97.79%)	ALA68779.1 (96.24%)
GE001981	618	70.05	8.76	SP(Sec/SPI)	WP_160269500.1 (99.03%)	ALA71875.1 (95.30%)
GE002130	437	49.26	4.05	SP(Sec/SPI)	WP_160270355.1 (28.57%)	ALA70136.1 (28.16%)
GE002405	381	42.86	9.47	LIPO(Sec/SPII)	WP_160269774.1 (99.48%)	ALA71443.1 (93.44%)
GE002430	391	43.35	8.92	SP(Sec/SPI)	WP_160269798.1 (93.61%)	ALA71420.1 (90.28%)
GE002476	168	18.34	10.27	SP(Sec/SPI)	WP_160269827.1 (98.81%)	ALA71363.1 (96.43%)
GE003057	335	37.00	5.89	SP(Sec/SPI)	WP_160270729.1 (97.31%)	ALA69727.1 (95.52%)
GE003085	98	10.36	9.26	SP(Sec/SPI)	WP_160270055.1 (98.98%)	ALA69758.1 (81.63%)
GE003250	329	37.49	9.17	SP(Sec/SPI)	WP_160270168.1 (96.66%)	ALA69899.1 (93.10%)
GE003377	432	47.72	6.06	SP(Sec/SPI)	WP_160270252.1 (92.13%)	ALA70034.1 (96.30%)
GE003445	134	15.01	9.96	SP(Sec/SPI)	WP_160270104.1 (76.00%)	-
GE003446	1338	145.36	9.07	SP(Sec/SPI)	WP_160270104.1 (38.24%)	ALA69775.1 (49.32%)
GE003450	452	49.14	9.67	SP(Sec/SPI)	WP_236658918.1 (66.67%)	ALA70130.1 (67.33%)
GE003532	1447	156.28	6	SP(Sec/SPI)	WP_160270104.1 (40.06%)	ALA69775.1 (52.21%)

## Data Availability

The whole genome sequence data reported in this paper have been deposited in the Genome Warehouse in the National Genomics Data Center, Beijing Institute of Genomics, Chinese Academy of Sciences/China National Center for Bioinformation, under accession number GWHBQTK01000000, which is publicly accessible at https://ngdc.cncb.ac.cn/gwh (accessed on 5 January 2023).

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
