# Peer review of "Comparative Genomic Analysis of a Thermophilic Protease-Producing Strain Geobacillus stearothermophilus H6"

_genes, 2023, doi:10.3390/genes14020466_

Round 1

Reviewer 1 Report

General comments

The research done in this study is very relevant for application in industry and very well executed. It is a comprehensive study that includes the complete pipeline of techniques from the start of isolating a relevant bacterial colony to the cloned end product of the enzyme under investigation and provides a very valuable manuscript for future authors. Therefore, I recommend that the authors give serious attention to the comments and rewrite the manuscript in terms of the following:

1.       The authors need to have a professional language editor relevant to the molecular field do language editing of the manuscript for it to be fit for publishing. In its current form, it is very difficult to follow.

2.       The materials and methods must be rewritten in the past tense.

3.       All the software used in the study must have references. Please indicate the versions of the software packages.

4.       The manufacturers of all kits used in the protocol need to be specified.

5.       The discussion is too short and can be expanded.

6.       The sequencing results and methodology with the relevant parameters must be made available as supplementary material.

Specific comments are included in the attached pdf file.

Author Response

Dear Reviewer,

First of all, we thank both reviewers and editor for their positive and constructive comments and suggestions.

Thank you very much for your comments and suggestions.

As suggested, we have made detailed and comprehensive changes to the manuscript, especially in the discussion section. We have revised the manuscript, according to the comments and suggestions of reviewers and editor, and responded, point by point to, the comments as listed below. Since the paper has been revised significantly throughout the text, we just highlight the crucial amendments in the revised manuscript, the small changes in the language on the text can be seen in review mode by Microsoft Word. The revised manuscript has been edited and proofread by English Language Editing Service. 

Response to Reviewer 1 Comments

Point 1: The authors need to have a professional language editor relevant to the molecular field do language editing of the manuscript for it to be fit for publishing. In its current form, it is very difficult to follow.

Answer:

The revised manuscript has been edited and proofread by English Language Editing Service.

Point 2: The materials and methods must be rewritten in the past tense.

Answer:

The materials and methods have been rewritten into the past tense and have been edited and proofread by English Language Editing Service.

Line 96-151

Point 3: All the software used in the study must have references. Please indicate the versions of the software packages.

Answer:

We have added references and versions of the software used in the study. And website of some software.

Line 96: MEGA 6.0

Line 101: Mauve 2.4.0

Line 110: Hifiasm v0.16.0 2

Line 113: Prodigal v2.6.3

Line 117: Circos v0.66

Line 126: SignalP v4.0

Line 127: TMHMM v2.0

Line 100: ANI (www.ezbiocloud.net/tools/ani).

Point 4: The manufacturers of all kits used in the protocol need to be specified.

Answer:

We have added manufacturers of all reagent kits used.

Line 85: the bacterial genomic DNA isolation kit (Mei5 Biotechnology Co. Ltd, Beijing, China)

Line 140: the gel recovery kit (Mei5 Biotechnology Co. Ltd, Beijing, China).

Point 5: The discussion is too short and can be expanded.

Answer:

We revised the discussion and added references.

Line 351-366: Added some discussion about Geobacillus have significant genetic diversity, and G. stearothermophilus H6 has the potential as a host for whole-cell applications and a biological tool in evolutionary engineering.

Line 391-402 Added some discussion about Geobacillus stearothermophilus can be the source of a variety of thermophilic enzymes and is widely used in a variety of biotechnology industries, and G. stearothermophilus H6 can produce protease with good activity at high temperature, which has a broad application prospect in biotechnology application.

Point 6: The sequencing results and methodology with the relevant parameters must be made available as supplementary material.

Answer:

The whole genome sequence data reported in this paper have been deposited in the Genome Warehouse in National Genomics Data Center, Beijing Institute of Genomics, Chinese Academy of Sciences / China National Center for Bioinformation, under accession number GWHBQTK01000000 that is publicly accessible at https://ngdc.cncb.ac.cn/gwh.

See the material method for the method. Line 109-122

Reviewer 2 Report

An interesting MS was written on a strain of Geobacillus stearothermophilus H6 that produces thermophilic proteases. I had enjoyed reading it. I would advise making a few modest changes to improve the standard.

1. The organism’s name needs to be in italics. There aren't many in the text and title either.

2. The abstract needs to be brief, with the main findings and a brief future outlook. Informing on the method is superfluous.

3. Fig. 2 and fig. 5 need to have better resolution.

4. Please provide a clear PCR image with accurate labeling (Fig 6).

5. The stain's ability to produce protease should be tested for temperature and pH stability.

6. Providing additional protease activity results is advised. For instance, variations in either fluorescence polarization (FP) or total fluorescence can be used to track the progress of FITC-casein substrate.

Author Response

Dear Reviewer,

First of all, we thank both reviewers and editor for their positive and constructive comments and suggestions.

Thank you very much for your comments and suggestions.

As suggested, we have made detailed and comprehensive changes to the manuscript, especially in the discussion section. We have revised the manuscript, according to the comments and suggestions of reviewers and editor, and responded, point by point to, the comments as listed below. Since the paper has been revised significantly throughout the text, we just highlight the crucial amendments in the revised manuscript, the small changes in the language on the text can be seen in review mode by Microsoft Word. The revised manuscript has been edited and proofread by English Language Editing Service.

Response to Reviewer 2 Comments

Point 1: The organism’s name needs to be in italics. There aren't many in the text and title either.

Answer:

We changed the organism’s name to italics.

Line 3: changed the “Geobacillus stearothermophilus H6” to“Geobacillus stearothermophilus H6”

Line 170: changed the “G. stearothermophilus H6” to“G. stearothermophilus H6”

Point 2: The abstract needs to be brief, with the main findings and a brief future outlook. Informing on the method is superfluous.

Answer:

We deleted the method in the summary.

Line 10-25: Geobacillus stearothermophilus H6 is an extreme thermophile Geobacillus strain isolated from hyperthermophilic composting at 80°C. Experiment showed that G. stearothermophilus H6 could produce extracellular protease with enzyme activity at 60 °C, which was much higher than that of Bacillus velezensis and Bacillus subtilis. In order to excavate the thermophilic enzyme in strain G. stearothermophilus H6, the whole genome of the strain was sequenced, and the gene function of the strain was predicted through genome annotation analysis. The G. stearothermophilus H6 draft genome consists of 3054993 bp, with a genome GC content of 51.66 %, and it was predicted to contain 3750 coding genes. The analysis showed that the strain H6 have a variety of enzyme coding genes, including protease, glycoside hydrolase, xylanase, amylase and lipase. The skimmed milk plate experiment showed that G. stearothermophilus H6 could produce extracellular protease to play a role at 60 °C, and the genome prediction contains 18 secreted proteases with signal peptides. These results showed that could provide a theoretical basis for the development and application of industrial strain.

Point 3: Fig. 2 and fig. 5 need to have better resolution.

Answer:

We change Figure 2 and Figure 5 to a higher resolution, Change the resolution to 600dpi

Line 205 and 271

Figure 2

Figure 5

Point 4: Please provide a clear PCR image with accurate labeling (Fig 6).

Answer:

We changed Figure 6 to a higher resolution and added more comments to the PCR image.

Line 333

Figure 6

Point 5: The stain's ability to produce protease should be tested for temperature and pH stability.

Answer:

Because the characteristics of the strains in this paper are heat-resistant strains, we use milk plate to compare the temperature of enzyme production of the strains in this paper. The research results show that the protease produced by G. stearothermophilus H6 has excellent temperature stability and can play a stable activity at 60 ℃, but the effect of pH on the enzyme production capacity has not been tested.

Point 6: Providing additional protease activity results is advised. For instance, variations in either fluorescence polarization (FP) or total fluorescence can be used to track the progress of FITC-casein substrate.

Answer:

We had used the skim milk plate experiment, which shows that the G. stearothermophilus H6 could produce protease and had activity. And we had used the bacterial solution to decompose casein substrate to detect protease activity, but the detection effect may not be ideal due to the low content of protease in the bacterial solution, and the experimental results could not be included in this paper. If you think it is necessary for us to add protease activity detection, we may design a new experiment to verify protease activity later.

Round 2

Reviewer 1 Report

Most of the questions and comments were answered in the revised version of the text. However, some of the questions are still unanswered:

Abstract:

The format of the abstract is still not correct. 

Line 11: Give a bit more general background on the applications of thermophilic strains, which would indicate the motivation for the study.

Line 11-14: Remove this sentence. It is repeated in line 22-23. Do not give a result before mentioning the methods employed. Please write one or two sentences to mention the methods employed broadly: e.g., “The strain was characterised with whole genome sequencing to identify putative extracellular enzymes.” Also mention comparison with databases to predict the sequence of the protease gene, as well as cloning and expression assays.

Introduction:

Line 30: Remove ‘to’

line 67: The last sentence of the introduction is actually a conclusion of the study results and should be removed. This sentence can be used in the abstract with great effect.

Line 111: How many reads? Please give the parameters used to filter the reads and the depth of sequencing.

Line 134: Replace “connected” with “inserted into”

Line 142-143: Please give the name of the kit with Vazyme as the reference for the manufacturer.

Line 326: correct “liquid culture”

Line 337: The sentence is incomplete.

Line 390: Remove “can”

Discussion:

The results of the study is not discussed in full and can still be improved.

Author Response

Response to Reviewer Comments

Abstract:

The format of the abstract is still not correct.

Line 11: Give a bit more general background on the applications of thermophilic strains, which would indicate the motivation for the study.

Line 11-14: Remove this sentence. It is repeated in line 22-23. Do not give a result before mentioning the methods employed. Please write one or two sentences to mention the methods employed broadly: e.g., “The strain was characterised with whole genome sequencing to identify putative extracellular enzymes.” Also mention comparison with databases to predict the sequence of the protease gene, as well as cloning and expression assays.

Answer:

Line 11: We added the application background of thermophilic strains.

Geobacillus are thermophilic gram-positive bacterium, which are widely distributed, and its ability to withstand high temperature makes it suitable for various applications in biotechnology and industrial production.

Line 13-15: We remove this sentence.

Line 16-18: We re-describe this sentence.

Through the whole genome sequencing and genome annotation analysis of the strain, the gene function of G. stearothermophilus H6 was predicted, and the thermophilic enzyme in the strain G. stearothermophilus H6 was mined.:

Line 25-27: We added the sequence cloning and heterologous expression analysis of protease gene.

By analyzing the sequence of the strain genome, a protease gene gs-sp1 was successfully screened. The gene sequence was analyzed and heterologously expressed, and the protease was successfully expressed in Escherichia coli.

Introduction:

Line 30: Remove ‘to’

line 67: The last sentence of the introduction is actually a conclusion of the study results and should be removed. This sentence can be used in the abstract with great effect.

The results are of great significance for mining the potential thermophilic protease and other enzymes of this strain and provide a basis for the application of Geobacillus stearothermophilus H6.

Line 111: How many reads? Please give the parameters used to filter the reads and the depth of sequencing.

Line 134: Replace “connected” with “inserted into”

Line 142-143: Please give the name of the kit with Vazyme as the reference for the manufacturer.

Line 326: correct “liquid culture”

Line 337: The sentence is incomplete.

Line 390: Remove “can”

Answer:

Line 33: Remove ‘to’

line 69-71: We removed the last sentence of the introduction.

Line 113: The original genome data was filtered and more than 2 kb of reads were retained.

Line 137: Changed the “connected” to “inserted into”

Line 145: Changed the “seamless cloning enzyme Vazyme” to “ClonExpress Ultra One Step Cloning Kit (Vazyme, Ninjing, China)”

Line 327: Changed the “culture liquid” to “liquid culture”

Line 339: We re-describe this sentence.

In this study, we isolated a strain of Geobacillus from hyperthermophilic compost and named Geobacillus stearothermophilus H6.

Line 398: Remove “can”

Discussion:

The results of the study is not discussed in full and can still be improved.

Answer:

We have added a discussion on the research results related to G. stearothermophilus H6 protease.

By analyzing G. stearothermophilus H6 genome protease, screened 18 proteases with signal peptide, selected gs-sp1 gene for heterologous expression, and successfully expressed the protease in E. coli BL21. Compared with homologous strain G. stearothermophilus B5 and G. stearothermophilus 10, the gs-sp1 protease had higher homology. Also, G. stearothermophilus H6 had its unique protease gene, such as GE003532 gene, which had low similarity among homologous strains. G. stearothermophilus H6 genome also contains a variety of other enzymes, which may have high thermal stability, and have broad application prospects in biotechnology applications.
